

# Regime shift of a large river as a response to Holocene climate change depends on land use – a numerical case study from the Chinese Loess Plateau

**Hao Chen[1, 2], Xianyan Wang[1*], Yanyan Yu[3], Huayu Lu[1] and Ronald Van Balen[2, 4*]**

[1]Frontiers Science Center for Critical Earth Material Cycling, School of Geography

and Ocean Science, Nanjing University, Nanjing 210023, China.

[2]Department of Earth Sciences, VU University Amsterdam, Amsterdam 1081HV, The

Netherlands.

[3]Key Laboratory of Cenozoic Geology and Environment, Institute of Geology and

Geophysics, Chinese Academy of Sciences, Beijing 100029, China.

[4]TNO-Geological Survey of the Netherlands

*Correspondence to*: Xianyan Wang (xianyanwang@nju.edu.cn), Ronald Van Balen

(r.t.van.balen@vu.nl)



**Abstract**

The Wei River catchment in the southern part of the Chinese Loess Plateau (CLP), is one of the centers of the agricultural revolution in China. The area has experienced

intense land use changes since ~6000 BCE, which makes it an ideal place to study the response of fluvial systems to anthropogenic land cover change (ALCC). We applied a numerical landscape evolution model that combines the Landlab landscape evolution model with an evapotranspiration model to investigate the direct and indirect effects of ALCC on hydrological and morphological processes in the Wei River catchment since

the mid-Holocene. The results show that ALCC not only leads to changes in discharge and sediment load in the catchment but also affects their sensitivity to climate change. When the proportion of agricultural land area exceeded 50% (around 1000 BCE), the sensitivities of discharge and sediment yield to climate change increased abruptly indicating a regime change in the fluvial catchment. It is associated with a large

sediment pulse in the lower reaches. The model simulation results also show a link between human settlement, ALCC and floodplain development: Changes in agricultural land use changes lead to downstream sediment accumulation and floodplain development, which in turn leads to further spatial expansion of agriculture and human settlement.

**Key words:** Anthropogenic land cover change, fluvial regime shift, fluvial climate sensitivity, climate change, Chinese Loess Plateau, central China



## 1 Introduction

Fluvial systems are affected by natural factors such as tectonics and climate change,

as well as by human activities (Bridgland, 2000; Bender et al., 2016; Best, 2019; Adams

et al., 2020; Bender et al., 2020; Li et al., 2020; Mossa and Chen, 2022). Many studies

have focused on the changes in river systems during the last century due to human-

activities, e.g. dams, reservoirs and mining (Vörösmarty et al., 2003; Brunier et al.,

2014; Zhao et al., 2015; Alfieri et al., 2017; Kong et al., 2017; Webber et al., 2017;

Zhang et al., 2021; Wang et al., 2022). However, human influence on the fluvial

catchments can date back thousands of years, to the beginning of the local agricultural

revolution (De Moor et al., 2008; Dotterweich, 2013; Ellis et al., 2016; Zhao et al.,

2022a).

Recent studies have raised the possibility that increasing anthropogenic stress on

a river system may increase its risk from future extreme climate events, e.g. floods and

droughts, since the resilience of the river system may be reduced, thus allowing it cross

a tipping point (Best and Darby, 2020; Choudhury et al., 2022). Determining the history

of a river can provide insight into the effects of anthropogenic perturbations that are

likely to unfold in its future evolution (Macklin and Lewin, 2019). However, the

response of the fluvial catchments to external perturbations is not straightforward

(Broothaerts et al., 2014; Guo et al., 2016; Verstraeten et al., 2017). Moreover, a regime

shift in a fluvial catchment is difficult to notice, since gradual changes caused by



external perturbations may alter the resilience of a fluvial catchment with only a small apparent effect on the current state (Scheffer et al., 2001). As a result, the extent to

which the vulnerability of fluvial catchments is affected by human-induced changes (e.g. land use) and the moment when the threshold could be crossed remains unclear. In addition, unraveling the mechanisms governing the response of a fluvial catchment to multiple, simultaneous forcings is notoriously difficult, especially in large river systems, where external factors and their effects are unique in each catchment and even

each river reach (Mao and Cherkauer, 2009; Fuller et al., 2015; Verstraeten et al., 2017; Macklin and Lewin, 2019). In consequence, a quantitative analysis of the interplay of the effects of climate change and land use change on the fluvial catchments is needed, and landscape evolution models (LEMs) provide a good opportunity for this (Zhao et al., 2022a,b).

The Chinese Loess Plateau (CLP) located in central China is affected by soil erosion seriously (Wang et al., 2006; Bloemendal et al., 2008; Zhao et al., 2013). Previous studies have already found that the erosion in the CLP is caused by both climate change and the development of agriculture which started a few thousand years ago (He et al., 2002; Huang et al., 2006; Chen et al., 2015; Chen et al., 2021). However,

the exact impact of these changes and the mechanisms involved remain largely unknown, especially for the time period around 1000 BC when a sediment pulse occurred in the river system (Song et al., 2020).

     The Wei River catchment, located in the southeastern part of the CLP, is one of the





most important sediment transport routes between the CLP and the Yellow River (Fig.1).

In this study, we combine the Landlab landscape evolution model (Hobley et al., 2017; Barnhart et al., 2020) with an evapotranspiration model (Thornton, 2010) to simulate the temporal and spatial changes of discharge and sediment yields in the Wei River catchment from 6000 BCE to AD 1850. In the simulations, we apply spatially and temporally varying precipitation and temperature, based on paleo-climate records

(Chen et al., 2015; Peterse et al., 2011). The simulated results from KK10 scenarios produced by Kaplan et al. (2011) are used to collect the changes of anthropogenic land cover. We specifically address the fluvial regime shifts in the Wei River catchment, which are reflected by changes in the sensitivity of discharge and sediment yield to climate change due to ALCC.


## 2 Study area

### 2.1 Geographic setting

As the largest tributary of the Yellow River, the Wei River is 818 km long and has a total drainage area about $1.35 \times 10^5$ km$^2$ (Guo et al., 2016) (Fig. 1a). The headwaters

of the river, which eventually drains into the Yellow River, are located in the Niaoshushan Mountains, in the western part of the CLP (Chang et al., 2016; Jia et al., 2021) (Fig. 1b). The Wei River catchment is located at the transitional zone of arid (north) to humid (south) areas. The catchment has an average annual precipitation of 500-700 mm and is a typical East Asian monsoon region. The precipitation mainly



occurs from June to September (Jia et al., 2021). The mean annual temperature ranges

from 7.8 ℃ to 13.5 ℃ (Tian et al., 2022). The mean annual discharge and sediment

load of the Wei River are $7.5\times10^9$ m$^3$ and $3.9\times10^8$ t (from 1956 to 2010), respectively

(Chang et al., 2016). The natural vegetation cover in the catchment changes from

deciduous broadleaf forest in the east to the temperate steppe in the west (Zhou et al.,

2015), and about 50% of the valley area is cultivated (Yu et al., 2016).

The northern part of the Wei River catchment is located in the southern part of the

CLP and is mainly covered by loess (Liu, 1985; Li and Lu, 2010). The tributaries

draining the CLP are relatively long and contribute large amounts of sediment (Fig. 1b)

(Chang et al., 2016; Jia et al., 2021). The southern part of the catchment lies in the

northern Qinling Mountains and has relatively short tributaries characterized by flash

flows (Fig. 1b) (Jia et al., 2021). The Wei River catchment has two major tributaries,

the Jing River and the Beiluo River (Fig. 1b). There are four types of landforms in the

catchment: 'hilly-gully', 'rocky-hill', 'table-gully' and 'fluvial-plain' areas. These

landforms also have different vegetation covers (Fig. 1c) (Yang, 2020). The 'hilly-gully

area' is located in the upper reaches of the main stream of the Wei River and in the

northern part of the catchment (Fig. 1c). The widely distributed steep gullies in these

areas cause a significant sediment yield (Chen et al., 2016; Zhang et al., 2020; Tian et

al., 2022). The 'rocky-hill area' includes the west-central and southern portions of the

catchment (Fig. 1c). The drainage divide between the Jing River and the Beiluo River

also belongs to this 'rocky-hill area' (Fig. 1c). It has a long history with high forest



cover (Zhang et al., 2017). The central parts of the Jing River and Beiluo River belong

to the 'table-gully area' (Fig. 1c), which is a high platform surrounded by gullies (Chen

et al., 2016). The middle and lower reaches of the main stream of the Wei River belong

to the 'fluvial-plain area' (Fig. 1c). They are mainly covered by alluvial deposits.


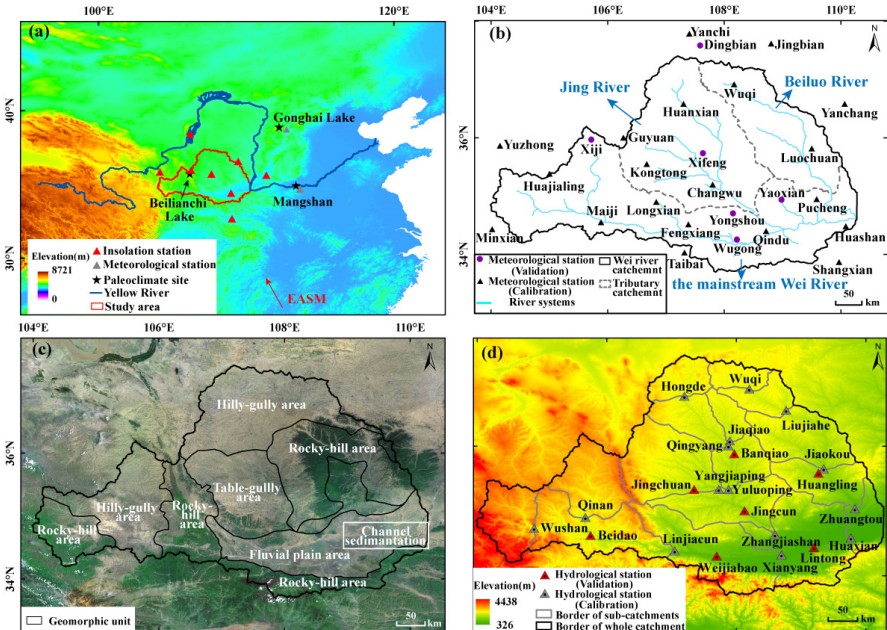

**Fig 1: The Wei River catchment a. Location of the Wei River and Yellow River; b. Meteorological stations and**

**rivers in and around the Wei River catchment; c. Landform types in the catchment; d. Hydrological stations**

**and sub-catchments (The topographic map used in fig. 1b,d is extracted from the NASA SRTM 90m digital**

**elevation model (https://srtm.csi.cgiar.org/); the satellite imagery used in fig. 1c is from © GoogleEarth map**

**(https://www.earthol.com/))**



2.2 Land-use history

In China, the Yellow River basin is one of the origins of agricultural civilization (Shen, 2000). In the Wei River catchment, numerous local agricultural cultures have developed since the mid-Holocene (Table 1). Agriculture first developed at about 6000 BCE (Li et al., 2009). It started with small settlements in the southeastern part of the catchment (Laoguantai Culture; Jia, 2003) or located on the terraces of tributaries in the northerwestern part of the catchment (Dadiwan Culture; Feng, 1985). Next, during the development of the Yangshao Culture, from 5000 BCE to 3000 BCE, the intensity of agriculture and the number of settlements increased (Li et al., 2009; Tan et al., 2011). Later, the Longshan Culture, from 3000 BCE to 2000 BCE, emerged in the Wei River catchment (Jin et al., 2002). The significant increase in charcoal concentration and the diversity of food utensils during this period indicate the rapid development of an agriculture-based civilization in the catchment area (Jin et al., 2002). From 2000 BCE to 1000 BCE, the food demand of the increasing population led to further expansion of the area of agricultural land (Zhao, 2004). In addition, more high-yield crops, such as wheat, were planted in this period (pre-Zhou and Western Zhou dynasty) (Zhao, 2004). After about 1000 BCE, the intensity of agricultural activity increased significantly, with more natural vegetation being converted to crop land due to innovations in agricultural technology (Jia, 2003). From about 1000 BCE to the present, the forest cover in the Loess Plateau decreased by about 44% (Zhao et al., 2013).





**Table 1 The development of agriculture in the Wei River catchment since the mid- Holocene**

**(Shi, 1986; Zhou, 2003; Yu et al., 2016)**

| Time period | Culture | Age |
|---|---|---|
| 6000 BCE – 5000 BCE | Laoguantai | 6000 BCE – 5000 BCE |
| | Dadiwan | 5850 BCE – 5400 BCE |
| 5000 BCE – 4000 BCE | Yangshao | 6800 BCE – 6000 BCE |
| 4000 BCE – 3000 BCE | Yangshao | 4000 BCE – 3000 BCE |
| | Majiayao | 3600 BCE – 2900 BCE |
| 3000 BCE – 2000 BCE | Majiayao | 2900 BCE – 2050 BCE |
| | Longshan | 2600 BCE – 2000 BCE |
| | Qijia | 2400 BCE – 1900 BCE |
| 2000 BCE – 1000 BCE | Siba | 1900 BCE – 1500 BCE |
| | Kayue | 1900 BCE – 1500 BCE |
| 1000 BCE – AD 1 | Xindian | 1600 BCE – 600 BCE |
| | Siwa | 1300 BCE – 500 BCE |


2.3 Hydrology and hydrological stations

The data from twenty-two hydrological stations are used (Fig. 1d). The Wushan, Qin'an, Beidao and Linjiacun hydrological stations are located in the upper reaches of the Wei River (Fig. 1d), where the mean annual discharge and sediment load account

for 26% and 30% of the entire catchment (Wang, 2013), respectively. Here, the discharge mostly originates between the Beidao and Linjiacun stations, while the sediment load is mostly produced in the upper area of the Qin'an station (Fig.1d) (Wang, 2013). The Weijiaobao, Xianyang, Lintong and Huaxian hydrological stations (Fig. 1d)



are located in the middle and lower reaches of the Wei River, which contribute about

48% of the discharge of the catchment (Zhang et al., 2007). The downstream part of the

Wei River, which includes the Lintong and Huaxian hydrological stations (Fig. 1d), is

a typical sediment accumulation area (Gao, 2006).

The Jing River is the largest tributary of the Wei River. About 71% of its sediment

is transported to the Wei River (Zhang et al., 2020). There are nine hydrological stations

located in this river: the Hongde, Jiaqiao, Qingyang, Yuluoping, Jingchuan, Yangjiaping,

Jingcun and the Zhangjiashan stations (Fig. 1d). The 73% of the discharge in the Jing

River catchment comes from the upper reaches of the Yangjiaping station and from the

reaches between the Yangjiaping, Yuluoping and Zhangjiashan stations (Zhang et al.,

2020). The sediment load mainly comes from upstream of the Yuluoping station,

accounting for 54% of the sediment load in the Jing River basin (Han, 2019).

For the Beiluo River, data from five hydrological stations are used in this work:

the Wuqi, Liujiahe, Jiaokou, Huangling and Zhuangtou stations (Fig. 1d). The 57% of

the discharge in the Beiluo River catchment is produced between the Liujiahe and

Zhuangtou stations (Ran et al., 2000). Most of the sediment load is produced in the

reaches upstream of the Liujiahe station, which accounts for 90.6% of the sediment load

in the Beiluo River basin (Zhang et al., 2017).





## 3 Materials and methods

### 3.1 Model development

In order to simulate the trend of fluvial sediment load and discharge under the impacts of land use and climate change, we apply the Landlab landscape model (Hobley et al., 2017; Barnhart et al., 2020) combining an evapotranspiration model (Thornton, 2010). The models are described in more detail in Chen et al.,(2021).

Simulations of the period from 1996 to 2016 are used to calibrate the model parameters by tuning the simulated discharges and sediment loads to the measured values at the hydrological stations. For simplicity, only the most important model parameters are calibrated, i.e. the effective root depth of plants and the erodibilities of the Base and Surface layers. The additional, less important parameters, such as biological parameters in the evapotranspiration model (Table S1) and the value of 'n' (a scaling exponent) in the Landlab's SPACE model component (Shobe et al., 2017) (Table S2), are provided by previous researches (Thornton, 2010; Shobe et al., 2017). Details of the calibration procedure are included in the Sect. 3.2. The calibrated model parameters are subsequently used in simulations for the time period from 6000 BCE to AD1850.

### 3.1.1 Evapotranspiration model

The simulated vegetation types in the Wei River catchment include deciduous broadleaf forest, grassland and crops, the distributions of which are shown in Sect. 3.3.



Ecological parameters of deciduous broadleaf forests and grasslands are based on the default values of the Biome-BGC model (White et al., 2000; Thornton, 2010). They have previously been successfully applied to the discharge and sediment load

simulations in one of the tributaries, the Beiluo River catchment (Chen et al., 2021).

For the crops, we use the winter wheat's ecological parameters since that is the dominant crop type in the Wei River catchment (Zhang et al., 1987). The soil nitrogen content, which is one of the required parameters in the model for the areas covered by crops, is assumed to be constant ($0.0004kgN/m^2$) to simulate the effects of fertilization

(Qin et al., 2010). The crop is irrigated twice during its growth and the timing of irrigation depends on local farming practices (Zhang et al., 1987). The applied value of irrigation each year is set equal to the mean annual value of irrigation in the Wei River catchment after the 1990s (Liu, 2003). In the modelling, similar to the previous studies (Hu et al., 2011), 80% of the stems and leaves are removed each year to simulate the

harvest processes.

### 3.1.2 Spatial distribution of climate data

For the calibration simulations of the period 1996 to 2016, we use the Kriging interpolation to compute the spatial distribution of evapotranspiration and runoff. The used meteorological data are the same as the previous simulations performed in the

Beiluo River catchment (Chen et al., 2021). These data, from twenty meteorological stations located in and around the catchment (Fig. 1b), are collected from the National



Meteorological Information Centre (http://data.cma.cn/). Eight insolation stations in and around the study area (Fig. 1a) are used to obtain the insolation data. We select another six meteorological stations (Fig. 1b) to test the accuracy of data obtained by

this method. The predicted and measured data match well ($R^2 > 70\%$, Fig. S1).

For the simulations during the Holocene (from 6000 BCE to AD 1850), the reconstructions of Holocene climate including precipitation and temperature (Peterse et al., 2011; Chen et al., 2015) are used to predict the climatic inputs, by methods of Chen et al. (2001). The predicted precipitation and air temperatures fit well with the

reconstructed data in Beilianchi lake (Zhang et al., 2020, 2021) (Text S2, Fig. S2), which is located in the northwestern part of the Wei River catchment (Fig. 1a). The additional data, i.e. Holocene atmospheric $CO_2$ concentration, are from the results of the Vostok ice core (Barnola et al., 1995; Petit et al., 1999), while humidity and sunshine duration are set equal to modern values. Additionally, the insolation values during

Holocene are calculated by the method of Laskar et al. (2004).

3.1.3 Anthropogenic land cover change

The changes of anthropogenic land use since the mid-Holocene (Fig. S3) is obtained from the KK10 database, which in turn is calculated from a global ALCC model that is driven by population density and the land suitability (Kaplan et al., 2009,

2011). The land suitability takes into account that agriculture develops first on the most productive crop lands (Kaplan et al., 2009). The simulated time series of land-use



change for the KK10 model is from 6000 BCE to AD 1850. Only the provincial data

from 221 BCE to AD 1850 are available to calibrate the spatial patterns of population

changes in China (Zhao and Xie, 1988). As a result, there is an uncertainty in the land-

use changes in our study region prior to 221 BC. In previous simulations focusing on

the Beiluo River catchment, Chen et al. (2021) applied a variation of 25% for the ALCC

from 6000 BCE to 221 BCE to estimate the impact of this uncertainty. It was shown to

have a limited effect on the simulation results for discharges and sediment loads.

3.1.4 Initial topography

There are two layers in the landscape evolution model, a Base layer and a Surface

layer (Shobe et al., 2017). The Surface layer consists of loose material, i.e. sediments,

and is above the Base layer, which is composed of bedrock. The initial topography for

the simulations is extracted from the NASA SRTM 90m digital elevation model (DEM)

(https://srtm.csi.cgiar.org/) and resampled to a spatial resolution of 1000 m. Since the

river network is disrupted after resampling, we resample the elevation of the network

separately and combine it with the previously resampled DEM. Then, the steady-state

topography is calculated over 5000 model years in order to remove DEM errors in the

fluvial network (e.g. Campforts et al., 2020; Sharma et al., 2021; Chen et al., 2021).

Subsequently, the elevation of the Baser layer used in the Holocene simulations is set

equal to the steady-state topography. Because we use similar erodibilities of loess and

sediment, the initial sediment thickness ( the thickness of the Surface layer ) is set to 0



m.

3.2 Calibration

Below, we present the calibrations of model parameters by fitting the simulated discharge and sediment load to the observed values at the hydrological stations (Fig. 1d). The discharge and sediment load data from another seven hydrological stations (Fig. 1d) are used for validation. Mean annual discharge and sediment load data measured at the stations are affected by e.g. dams and irrigation systems. These data were corrected into natural discharge and sediment load data using the method by

Chang et al.(2016). Details are presented in the supplemental materials (Text S1, Table S3). The calibrated parameters are the effective root depth of plants and the erodibilities of the Base and Surface layers. The calibration results are accepted when the mismatch between the simulated and observed discharges and sediment loads is less than 10%.

3.2.1 Effective root depth of plants

For the evapotranspiration model, we only calibrate the effective root depth, which is determined by the vegetation types and soil environment (Vörösmarty et al., 1989), because it has the largest impact on the evapotranspiration rates and soil water content in the evapotranspiration model.

    For the root depth calibration, the catchment is subdivided into sixteen sub-

catchments (Fig 1d). We use the present-day precipitation and temperature, and fit the mean annual discharge at the outlet of each sub-catchment (gray triangle in Fig. 1d).



The initial effective root depths of plants are set at 1.5 m for deciduous broadleaf forest

and 1m for grass and crop lands, based on the average root depth in the Loess Plateau

(You et al., 2009). During the iterative calibration processes we vary the root depth

incrementally by 1 cm, while the difference between the initial root depths of trees and

grass/croplands is kept constant. After calibration, the mismatches between the

observed and simulated annual discharge at the calibration stations are between 0.48%

and 6.10% (Fig. S4).

In order to validate the accuracy of our calibrated results, we further compare the

differences between the predicted and natural annual discharge at another seven

hydrological stations (red triangle in Fig. 1d). The results show that the model predicts

annual discharge with an error that ranges between 0.35% and 7.95% (Fig. S4).

3.2.2 Erodibilities

For the parameters in the Landlab model, we calibrate the erodibilities of the Base

and Surface layers based on measured annual sediment loads (Fig. 1d). The initial

values of the Base layer's erodibility are calculated based on the geologic map (Fig

S5a), which is extracted from a 1:250,000 digital geologic map of China (Zuo et al.,

2018), and the parameters of bedrock's erodibility from the LAPSUS model (Schoorl

and Veldkamp, 2001). For the initial values of the Surface layer's erodibility, we

calculate the values by the method of Hancock et al. (2019), which uses the soil

properties and the NDVI data (Normalized Difference Vegetation Index). The data of



soil properties (Fig. S5b-e) come from the China soil map, which in turn is collected

from the Harmonized World Soil Database (v1.1) (Nachtergaele et al., 2010). The

NDVI data (Fig. S5f) are based on the SPOT vegetation index database of China

(http://westdc.westgis. ac.cn/).

Then, the Base layer's and Surface layer's erodibilities in each sub-catchment are

adjusted until the simulated sediment load matches the observed data at the hydrological

stations located at the sub-catchment's outlet (gray triangle in Fig. 1d). After

calibrations, the mismatches between the observed and simulated annual sediment load

at the calibration stations range from 0.01% to 9.39% (Fig. S6). For the validation

stations, the model predicts annual sediment load with an error that ranges between

1.50% and 7.27% (Fig. S6).

3.3 Holocene simulations

Two model scenarios (*a model with land use and climate change, Normal, and a*

*model without climate change, WCC*) are used in the Holocene simulations. *Scenario*

*Normal* uses reconstructed paleo-climate data and KK10 land-use data to model the

spatial and temporal changes in water and sediment discharges due to climate change

and anthropogenic land cover changes. *Scenario WCC* is used to study solely the effects

of land use change; the climatic conditions are kept constant.

In order to demonstrate the effects of anthropogenic land cover change on drainage

hydrology, we calculate the changes of discharge and sediment yield at the outlet as





well as their coefficient of spatial variation (CV, Eq. (1)) for the whole catchment.

$$CV = \frac{1}{x_a} \sqrt{\frac{\sum_{i=1}^{n}(x_i - x_a)^2}{n}} \qquad (1)$$

Where, $CV$ is the coefficient of spatial variation of the mean annual discharge or

sediment yield. $x_a$ is the average discharge or sediment yield of the whole catchment. $x_i$

is the discharge or sediment yield in the $i$ grid-cell, and $n$ is the total number of grid-

cells.

Next, the sensitivities of the mean annual discharge and sediment yield as well as

their spatial variation coefficients to the climate change are calculated based on the

differences between the *Normal* and *WCC* scenarios (Eq.(2)). Finally, the calculated

sensitivities are correlated with the different intensities of human activity to reveal the

impact of land use change on the fluvial response mechanisms.

$$S_c = \left| \frac{dif_{climate-basic}}{\Delta climate} \right| \qquad (2)$$

Where, $S_c$ is the sensitivity of the simulation results to climate change. A higher

value of $S_c$ means a more sensitive response. $dif_{climate-basic}$ is the difference between the

simulation results between the *Normal* and *WCC* scenarios. The simulation results are

the mean annual discharge, the CV of the mean annual discharge, the mean annual

sediment yield and the CV of the mean annual sediment yield, resulting in four different

sensitivity values. $\Delta climate$ is the difference between the climate conditions for the

scenarios. We use the difference of precipitation between both scenarios as the

$\Delta climate$ parameter, since precipitation is the climate parameter that has the most

significant impact on the simulation results (Chen et al., 2021).



In the Holocene simulation, the time-step is one year. For simplicity, the annual climatic parameters and anthropogenic land cover are set constant for each 500 years.

For the modeling of discharge and sediment load, the distributions of natural plants are determined by the pollen-based reconstruction of main vegetation types in different geomorphic units in the Loess Plateau (Sun et al., 2017). By associating the reconstructed vegetation data with the modern geomorphic distribution map (http://www.geodata.cn), the natural plants in the Wei River catchment are divided into

forest and grass (Fig. S7).

**4 Results**

4.1 *Normal scenario*

4.1.1 Runoff and discharge

The evolution of the simulated runoff and discharge from 6000 BCE to AD 1850

is shown in Fig. 2. The mean annual runoff (catchment average) increases about 16.5% during the first 2000 years (6000 BCE to 4000 BCE). A second increase of around 6.7% takes place from 2000 BCE to 500 BCE (Fig. 2A). Spatially, the simulated runoff rates show a gradual increasing trend from north to south, which is caused by the distribution of mean annual precipitation. A low value occurs in the middle reaches of

the Jing River and the Beiluo River, which may be caused by the high value of effective root depth (Fig. S4) causing a locally high value of evapotranspiration.

In each sub-catchment, the fluctuations of the mean annual discharge are similar





(Fig. 2B). The main contribution to the discharge at the catchment outlet is provided

by the downstream part of the Wei River (Fig. 2b1). During the Holocene, the

discharge decreases by 13.2%, 26.7%, 20.7%, 31.1%, 35.1% and 21.8% in the Wushan,

Qin'an, Linjiacun, Xianyang, Huaxian and Outlet sub-catchment, respectively (Fig.

2b1). In the Jing River, the discharge decreases by 37.3%, 20.2%, 37.3%, 44.7%, 22.7%

and 47.2% in the Hongde, Jiaoqiao, Qinyang, Yuluoping, Yangjiaping and

Zhangjiashan sub-catchment, respectively (Fig. 2b2). In the Beiluo river, reductions

about 33.1%, 29.6%, 39.9% and 50.4%, in the Wuqi, Liujiahe, Jiaokou and Zhuangtou

sub-catchment, respectively (Fig.2b3).

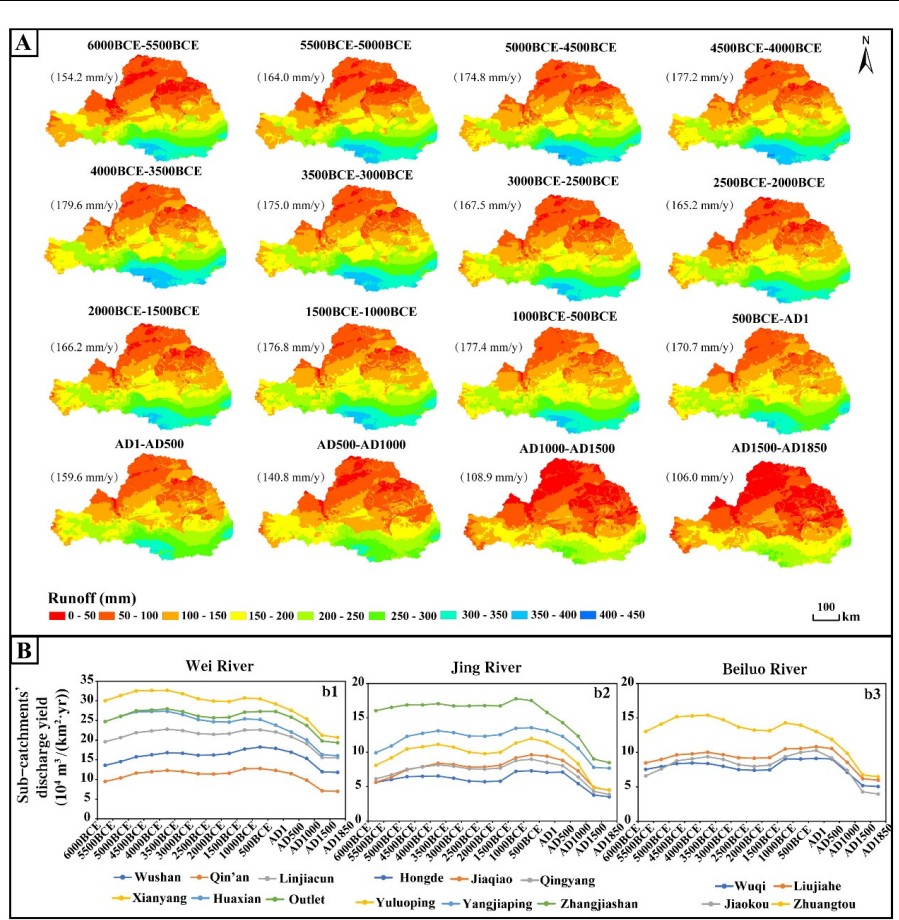

Fig 2: Simulated mean annual runoff (A) and the time-trend of sub-catchment mean annual discharge (B).

### 4.1.2 Sediment thickness and sediment yield

Figure 3 shows the distribution of sediment thickness and the evolution of sediment yield in each sub-catchment. The sediment thickness has a decreased trend from northwest to southeast. The upper and lower reaches of the main stream of the Wei River have a thin accumulation of sediment (less than 2 m) (Fig. 3A). A prominent sediment accumulation is predicted in the lower reaches from 1000 BCE onwards; its

lateral extension results from lateral channel migration (Fig. 3A).



For the main stream of the Wei River, the sediment flux mainly comes from the Qin'an sub-catchment (Fig. 3b1). There is no sediment yield at the outlet of the Wei River, which indicates it is a sedimentation zone (Fig. 3b1). Sediment yields are higher in the Jing River than in other sub-catchments; the maximum value is located in the

Yuluoping sub-catchment (Fig. 3b2). In the Beiluo River, the sediment is mostly produced in the Wuqi and Liujiahe sub-catchments (Fig. 3b3). The trends of mean annual sediment yield in each sub-catchment are similar and had a total increase 10 to 30 times during the simulation (Fig. 3B). Before 1000 BC, the mean annual sediment yield had an approximately steady, linear increase in all sub-catchments (Fig. 3B).

Subsequently, they experienced a sharp increase between 1000 BC and AD1. Then, a rapid decrease occurred after AD 1 (Fig. 3B).





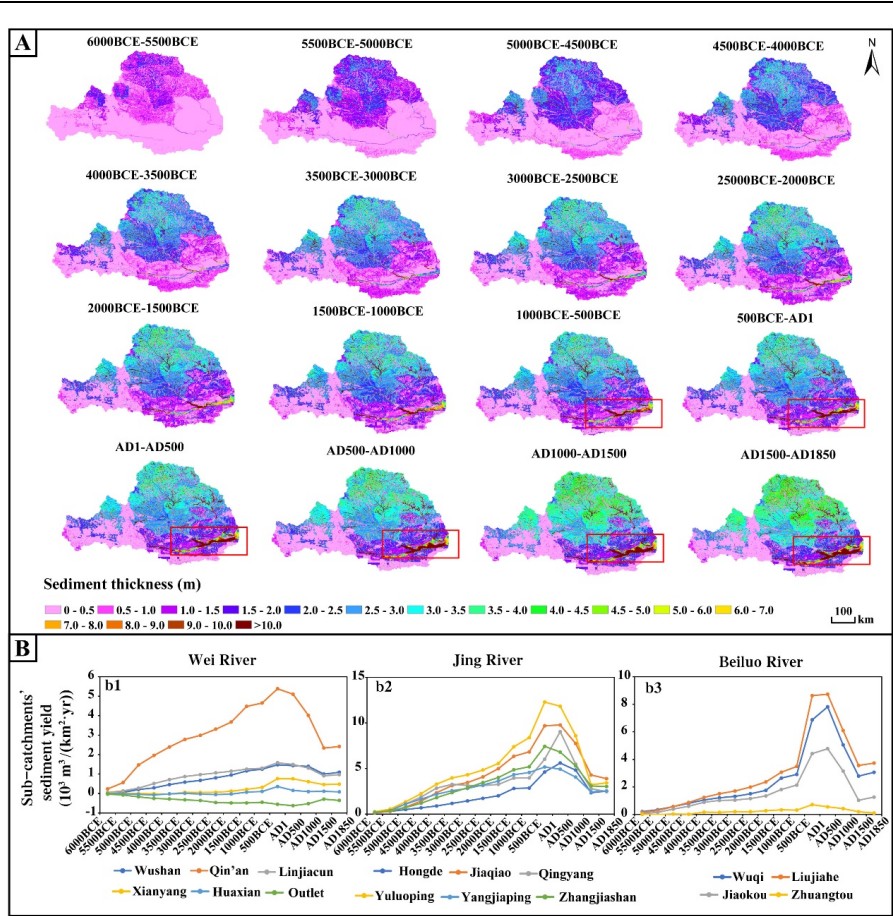

Fig 3: Sediment thickness (A) and the evolution of sub-catchment mean annual sediment yield (B)

4.2 The difference of model results for the two scenarios

The spatial distributions of mean annual runoff in the *Normal* and *WCC* scenarios are similar (Figs. 2A, S8A). However, the evolution of simulated mean annual runoff (catchment average) has fluctuations in the *Normal* scenario while it is almost linearly increasing in the *WCC* scenario (Figs. 2A, S8A). Compared to *Normal* scenario (Fig.



2B), the discharge yields of the sub-catchments belonging to the southeastern part (such

as Zhangjiashan and Zhuangtou) increase appreciably in the *WCC* scenario (Fig. S8B).

The comparison of the mean annual discharge (catchment average) and its spatial

variation coefficient for these two scenarios show that the impacts of climate change on

the temporal and spatial trends of discharge start to increase after AD 1 (more than 20%)

(Fig 4a, b).

In both the *Normal* and *WCC* scenarios, a large accumulation occurs in the middle

reaches of Jing River at about 4000 BCE and in the downstream part of the Wei River

around 1000 BCE (Figs. 3A, S9A). However, the sediment thickness in the northern

part of the catchment is thicker in the *Normal* scenario than in the *WCC* scenario (Figs.

3A, S9A). In the *WCC* scenario (Fig. S9B), the sediment yields of the sub-catchments

located in the northwestern part (such as Jiaoqiao, Qinyang) are larger compared to the

*Normal* scenario (Fig. 3B). Based on the comparison of the results of the *Normal*

scenario to those of the *WCC* scenario, the intensity of the impact of climate change on

the mean annual sediment yield (in the *Normal* scenario) increases to some extent after

AD 1 (more than 20%, Fig 4c). The spatial variation coefficient of sediment yield in

the two scenarios are almost the same during the simulation (less than 20%, Fig 4d),

which indicates that land use change is the dominant factor for the spatial characteristics

of sediment yield.



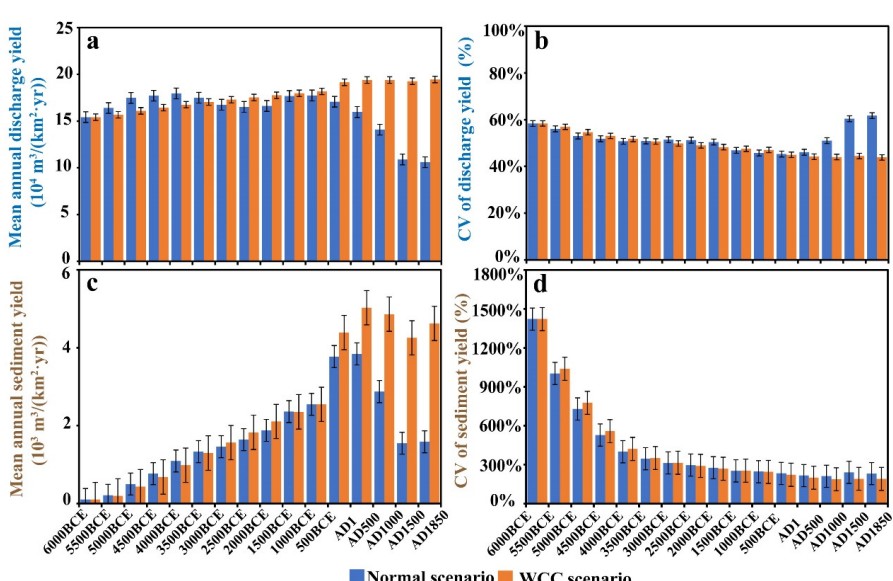

**Fig 4: Comparison of mean annual discharge yield (a), CV of discharge yield (b), mean annual sediment yield (c) and CV of sediment yield (d) for the *Normal* and *WCC* Scenarios**

## 5 Discussion

5.1 A regime shift around 1000 BC

The sediment thickness distribution in the *Normal* scenario shows a significant increase in the lower reaches of the main Wei River after 1000 BCE (Fig. 3A, red rectangle). The mean annual sediment yield in each sub-catchment also experiences a large increase at the same time (Fig. 3B). These changes are not only a consequence of the large increase of the land use around the 1000 BC (Chen et al., 2021), but also a signal indicating the change of sensitivity of the fluvial catchment to climate change as a result of increasing ALCC.





The sensitivity to climate change of mean annual discharge and sediment yield and their coefficient of spatial variation alters abruptly when the areal extent of land use exceeds a certain threshold (Fig. 5). When the areal extent is low (<30 %), the

sensitivity to climate change declines steadily with increasing areal extent (and thus also with time; Fig. 5). The sensitivity fluctuates when the areal extent is between 30% and 50% (Fig. 5). However, when the areal extent of land use is high (>50%), starting at ~1000 BCE, the sensitivity increases with increasing areal extent (Fig. 5) indicating a regime shift of the fluvial catchment.

These changes in sensitivity are associated with the areal extent of land use and the type of vegetation change. In the catchment, the natural vegetation is made up of forest and grass (Fig. S7), which is converted to cropland. Runoff in grassland is more sensitive to climate change than runoff in cropland, whereas runoff in forest is less sensitive than runoff in cropland (Mao and Cherkauer, 2009). The main vegetation

change in the catchment during the time period from 6000 BCE to around 3000 BCE is from grass to the crop in the western and northern parts of the catchment (Fig. S3), which leads to the decreased sensitivities of the catchment discharge and sediment yield and their spatial variations, CV, to climate change (Fig. 5a, b). From around 3000 BCE to 1000 BCE, the sensitivities first sharply increase and then decrease rapidly, which

shows their instability. This may be caused by changes of other parameters, like air temperature (Fig. S2), which are not considered. From 1000 BCE onwards, the major anthropogenic vegetation changes are from forest to crop in the southeastern part of the




catchment (Fig. S3), which results in the increase of the sensitivities (Fig. 5).

Sediment accumulation that first occurs in the middle reaches of the Jing River

and the Beiluo River at ~ 4000 BCE, and subsequently migrates to the lower reaches of

the main Wei River at ~ 1000 BCE, indicates a sediment wave, which has also been

reported in other catchments (Van Balen et al., 2010; James and Lecce, 2013). Since

the significant aggradation in the lower reaches of the main Wei River at ~ 1000 BCE

reflects a regime shift of the whole fluvial catchment, the earlier sediment accumulation

in the middle reaches of the Jing River and the Beiluo River at ~ 4000 BCE maybe

indicate an earlier regime shift in the tributaries.

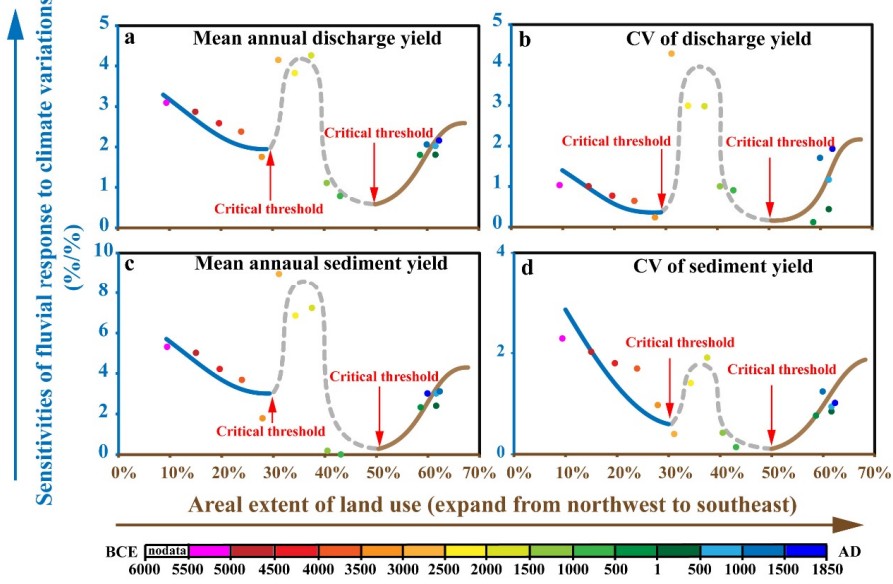

**Fig 5: Sensitivity of discharge and sediment yield to the climate changes due to increasing areal extent of land**

**use with time. The colors of points indicate the simulated time; 50% land use area corresponds to 1000 BCE.**



5.2 Coupling between ALCC by human settlement and floodplain development

The spatial distribution of sediment accumulation correlates with the distribution of archaeological sites since the mid-Holocene (Yu et al., 2016; Fig. 6). Sediment accumulation and floodplain construction first occurs in the downstream part, then expands to the northwestern part and becomes concentrated in the upstream part in the

western part of the CLP, in the Qi'an and Yangjiaping sub-catchments (Fig. 6a). This pattern of expansion is consistent with the spatial trend of the growth of the number of archaeological sites during the same period (Yu et al., 2016). This can be explained by the spatially asynchronous development of floodplains, caused by migration of sediment waves in the catchment. The floodplains provided ideal locations for initial

settlements (Clevis et al., 2006). These new settlements, in turn, would have led to an increase in local land use, which in turn, would result in higher sediment yields. These sediments are then transported further and accumulated downstream, which results in floodplain development there and thus provides new suitable places for further human settlement (Fig. 7). This resembles the niche construction theory (NCT) from biological

and ecological systems (Laland et al., 1996; Laland et al., 1999; Laland et al., 2001; O'Brien and Laland, 2012). The NCT places emphasis on the capacity of organisms, in this case humans, to modify their environment and thereby act as co-directors of their own, and other species' evolution (Laland et al., 2001; Spengler, 2021).



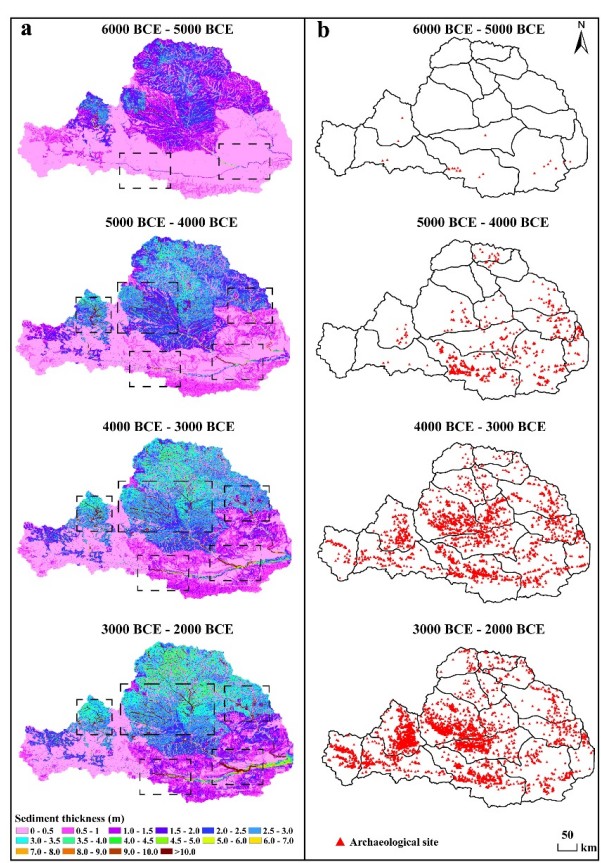

**485**  Fig 6: The spatial correspondence between the location of sediment aggradation (a) and archaeological sites

(b, Yu et al., 2016) during the mid-Holocene.



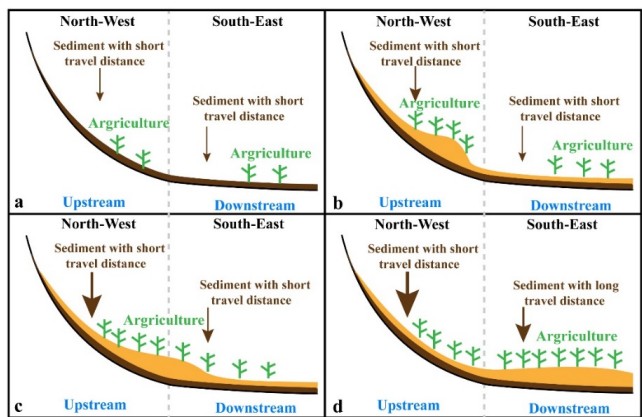

**Fig 7: Conceptual model for the relationship between expansion of agriculture and fluvial floodplain**

**aggradation.**

## 6 Conclusions

Land use change in the Wei River catchment in the Chinese Loess Plateau (CLP) not only has a significant direct impact on discharge and sediment yield, but also alters

the resilience of this fluvial catchment to climate change. The sensitivity of the entire catchment to climate change decreases with increasing amounts of areal extent of land use, as long as it is less than 30%, when the increase in areal extent is dominated by a change from grass to cropland. The sensitivity increases when the areal extent of land use exceeds 50% of the catchment around 1000 BCE. During this increase in extent,

the main vegetation cover changed from trees to crops. This regime shift can be reflected by the significant sediment accumulation in the lower reaches starting around 1000 BCE. There is an initial sediment aggradation in the middle reaches of Jing River and Beiluo River (~ 4000 BCE), which migrates to the lower part of Wei River around





3000 years ago. This migration of sedimentary waves suggests that the regime shifts

occur earlier upstream in the tributaries. Our simulation results also suggest that there

is a coupling between early land use, sediment accumulation and resultant floodplain

development: new settlements on floodplains lead to further increases in sediment yield

and floodplain formation further downstream.

**Code and data availability**

The mapped shapefiles and the code used to process the mapped data are available

upon request to the corresponding author. The data for the KK10 scenario can be

accessed at https://doi.pangaea.de/10.1594/PANGAEA.871369. The documentation of

the Landlab can be found at https://landlab.readthedocs.io/ and the newest version of

the software is archived at https://doi.org/10.5281/zenodo.3647556 and https://

doi.org/10.5281/zenodo.3644240. The source of Biome-BGC model can be accessed at

http://www.ntsg.umt.edu.

**Author contribution**

Hao Chen wrote the code and performed the simulations. Xianyan Wang and

Ronald van Balen modified the code and improved the simulations. Yanyan Yu provided

the data of archaeological sites. Huayu Lu modified the main text of the manuscript.

Hao Chen wrote the manuscript with contributions from all co-authors.

**Competing interests**

The authors declare that they have no conflict of interest.



**Acknowledgements**

We appreciate Professor Jed O Kaplan for providing the anthropogenic land cover

change data simulated by the KK10 model. We also thank Professor Kuang xueyuan

for the simulated paleo-climate results from CESM model. This research is supported

by the National Natural Science Foundation of China (42021001, 41971005), Second

Tibetan Plateau Scientific Expedition Program (2019QZKK0205).

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
