# Peer review of "Past anthropogenic land use change caused a regime shift of the fluvial response to Holocene climate change in the Chinese Loess Plateau"

_EGUsphere, 2023_

## Author Comment (AC1)

**Past anthropogenic land use change caused a regime shift of the fluvial response to Holocene climate change in the Chinese Loess Plateau**

**Hao Chen[1, 2], Xianyan Wang[1*], Yanyan Yu[3], Huayu Lu[1] and Ronald Van Balen[2, 4*]**

[1]Frontiers Science Center for Critical Earth Material Cycling, School of Geography and Ocean Science, Nanjing University, Nanjing 210023, China.

[2]Department of Earth Sciences, VU University Amsterdam, Amsterdam 1081HV, The Netherlands.

[3]Key Laboratory of Cenozoic Geology and Environment, Institute of Geology and Geophysics, Chinese Academy of Sciences, Beijing 100029, China.

[4]TNO-Geological Survey of the Netherlands

*Correspondence to*: Xianyan Wang ([xianyanwang@nju.edu.cn](mailto:xianyanwang@nju.edu.cn)), Ronald Van Balen ([r.t.van.balen@vu.nl](mailto:r.t.van.balen@vu.nl))

**Summary:**

The authors would like to thank the Reviewer for the constructive and insightful comments on our manuscript, which were helpful to significantly improve the manuscript and its readability.

We carefully revised the manuscript following the Reviewer's suggestions, in order to better highlight the impact of past land use change on the sensitivity of the Wei River catchment to climate change. We have redesigned the title of the manuscript, and also reorganized the Introduction to make the objective of this work clearer. We also modified the Discussion and Conclusions to show when and in what manner land use change has affected the river's sensitivity to climate change.

Moreover, we have significantly expanded the Materials and Methods section, as

suggested by the Reviewer, to make readers easier understand our simulation approach. This expansion includes a technology roadmap of the landscape evolution model, the methods to obtain the paleoclimate data, as well as the temporal and spatial resolutions of model outputs. The uncertainties about the paleoclimate data, model parameters, model resolution, and initial topography in 6000 BCE are also explained and evaluated in detail in the main text to make our simulation results more convincible.

Overall, in the new version of the manuscript, the comments and suggestions raised by Reviewer are fully considered. We think the modified paper shows interesting behavior of the Wei River catchment, which has general implications for other river systems. We think the modified manuscript can meet the reviewer's expectations.

In the following, we discuss in detail all the Reviewer's comments and show how we have addressed them in the revised manuscript. Please note that the Reviewer's comments are in black, our responses are in blue, and the content of the revised manuscript is depicted in a frame.

**Legend**

RC: Reviewer Comment;    AR: Author Response;    □: Modified manuscript content

**Response to the Reviewers #1:**

RC 1: The numerical modeling is the core method. However, it is not introduced enough in Introduction. Applications of landscape evolution modeling for other similar studies should be mentioned.

AR 1: Thanks for your comment and suggestion. A statement about landscape evolution modeling for other similar studies has been added in the main text (line 69-78).

(line 69-78) Therefore, landscape evolution models (LEMs) have been widely used to investigate the development of fluvial morphology under the impacts of external disturbance (Tucker and Hancock, 2010; Van Balen et al., 2010; Coulthard and Van de Wiel, 2013; Pan et al., 2021; Zhao et al., 2022a,b). They have been used to study the influence of vegetation (Istanbulluoglu and Bras, 2005; Carriere et al., 2019), climate (Routschek et al., 2014; Manley et al., 2020), and their combined effects (Schmid et al., 2018; Sharma et al., 2021). For example, Sharma et al. (2021) found that the effect of Milankovitch periodicity variations on erosion is lower in sparsely vegetated landscape than in densely vegetated landscape. However, the changes of fluvial response to climatic variations caused by land use change still needs further study.

RC 2: Line 85: what is "KK10 scenarios" in this sentence? We do not know anything about KK10 before the line. You should briefly describe what is KK10.

AR 2: Thanks for your comment and suggestion. A brief description about the KK10 scenarios has been added in the main text (line 84-87).

(line 84-87) The changes of anthropogenic land use are taken from Kaplan et al. (2010). Their KK10 database provides the anthropogenic land cover change from 8000 years ago to AD 1850, based on a model that relates changes of global population to past land use (Kaplan et al., 2009).

RC 3: Line 99: the study area is not an East Asian monsoon region but just belongs to it.

AR 3: Thanks for your corrections. We have changed the sentence to the "The catchment has an average annual precipitation of 500-700 mm and belongs to the East Asian monsoon region (line 97-99)."

RC 4: Line 157: what is the time scale? daily, monthly, or yearly? what is the time range of observed data?

AR 4: Thanks for your comment. The observed discharge and sediment load are yearly data. The new statement about this has been added in the main text (line 153).

The time ranges of available discharge and sediment load data are different in each hydrological station. The statement about the time range for each hydrological station has been added in the main text (line 153-157; line 159-162; line 165-166; line 168-171; line 171-173; line 175-177; line 177-180).

(line 153) The yearly data from twenty-two hydrological stations are used (Fig. 1c).

(line 153-157) The Wushan, Qin'an, Beidao and Linjiacun hydrological stations are located in the upper reaches of the Wei River (Fig. 1c), where the mean annual discharge and sediment load account for 26% and 30% of the entire catchment (from 1956 to 2000; Wang, 2013), respectively.

(line159-162) The Weijiaobao, Xianyang, Lintong and Huaxian hydrological stations (Fig. 1c) are located in the middle and lower reaches of the Wei River, which contribute about 48% of the discharge of the catchment (from 1956 to 2000; Zhang et al., 2007).

(line 165-166) About 71% of its sediment is transported to the Wei River (from 1956 to 2015; Zhang et al., 2020).

(line168-171) 73% of the discharge in the Jing River catchment comes from the upper reaches of the Yangjiaping station and from the reaches between the Yangjiaping, Yuluoping and Zhangjiashan stations (from 1956 to 2015; Zhang et al., 2020).

(line 171-173) The sediment load mainly comes from upstream of the Yuluoping station, accounting for 54% of the sediment load in the Jing River basin (from 1959 to 2016; Han, 2019).

(line 175-177) 57% of the discharge in the Beiluo River catchment is produced between the Liujiahe and Zhuangtou stations (from 1957 to 2009; Ran et al., 2000, 2012).

(line 177-180) Most of the sediment load is produced in the reaches upstream of the Liujiahe station, which accounts for 90.6% of the sediment load in the Beiluo River basin (from 1957 to 2009; Zhang et al., 2017).

RC 5: Line 183-198: a technology roadmap of model development should be given here.

AR 5: Thanks for your comment. A technology roadmap of the landscape evolution model has been added in the main text (line 188).

(line188; Fig 2: Technology roadmap of the landscape evolution model.)

[Figure]

RC 6: Line 189: please describe what the tuning method looks like.

AR 6: Thanks for your comment. A brief description about the tuning method has been added in the main text (line 192-195).

(line 192-195) The tuning method is an iterative calibration process that changing the parameter from an initial value to the most appropriate value to minimize the mismatch between the simulated and observed hydrological data.

RC 7: Line 225: in the Fig S1, sample numbers for validation may be too small, which means the high R2 cannot pass the significance test. It may bring more uncertainties into the following results.

AR 7: Thanks for your comment. Because of the limited number of meteorological stations (total numbers are twenty-six) in the study area, the sample numbers for validation were small. In order to guarantee the accuracy of the results from the Kriging interpolation, we used twenty meteorological stations located in and around the catchment (Fig. 1c) to do the interpolation. Then, we used another six meteorological stations (Fig. 1c) to do the validation.

We also did the significance test for these validated results by calculating the P-Value. The P-Values are all lower than 0.01, which means the high $R^2$ could pass the significance test. We have added the calculated P-Values in the Fig S1.

(Supplemental Materials; Fig S1: Scatter plot of predicted and observed meteorological data at the validation stations in Wei River catchment)

[Figure]

RC 8: Line 226-229: "the reconstructions of Holocene climate including precipitation and temperature (Peterse et al., 2011; Chen et al., 2015) are used to predict the climatic inputs, by methods of Chen et al. (2001)." makes me confused. Did you use Holocene precipitation and temperature to predict climatic inputs? What do climatic inputs mean here? What is the method of Chen et al. (2001)? In addition, the word "predicted" is inappropriate, it may be some words like "simulated" because it is nothing about the future.

AR 8: Thanks for your comments and suggestions. We used the pollen-based reconstruction of annual precipitation from Gonghai Lake (Chen et al., 2015) to calculate the Holocene precipitation time series, and the reconstruction of air temperature from a loess-paleosol sequence at the Mangshan loess plateau (Peterse et al., 2011) was used to calculate the Holocene temperature time series. The calculation method is the same as that used in Chen et al. (2021). We modified the statement about these in the main text to make the description clearer (line 232-235). We also replaced the word "predicted" with "calculated" in the main text, as you suggest (line 235-238).

(line 232-235) The Holocene precipitation and temperature series are calculated based on the annual precipitation reconstruction from Gonghai Lake (Chen et al., 2015) and the air temperature reconstruction from the Mangshan Loess plateau (Peterse et al., 2011). The methods are the same as those used in Chen et al. (2021). (line 235-238) The calculated precipitation and air temperatures fit well with the reconstructed data from Beilianchi lake (Zhang et al., 2020, 2021) (Text S2, Fig. S2), which is located in the northwestern part of the Wei River catchment (Fig. 1a).

RC 9: Line 238: how about the uncertainty of KK10 database calculated from a global ALCC model for the local catchment in this study?

AR 9: Thanks for your comment. We applied a variation of 25% for the ALCC from 6000 BCE to 221 BCE to estimate the impact of the uncertainty of the KK10 database on the discharge and sediment load predictions in a tributary, the Beiluo river catchment (Chen et al., 2021). The results showed that variations of 25 % of the ALCC (i.e. KK10) has a little effect (line 253-257). Since the Beiluo river is a tributary of the Wei River, we consider the impact of the uncertainty of the KK10 database to be similarly small.

(line 253-257) In previous simulations focusing on a tributary, the Beiluo River catchment, Chen et al. (2021) applied a variation of 25% for the ALCC from 6000 BCE to 221 BCE to estimate the impact of this uncertainty. It showed the uncertainty of ALCC had a limited effect on the simulation results for discharges and sediment loads.

RC 10: Line255-260: how did authors guarantee the reasonability of initial topography? If not, the study may be inauthentic and become a sensitivity test.

AR 10: Thanks for your comment. We used the modern topography to calculate the initial topography that we applied in the Holocene simulations. Recent studies have proved that the modern topography can be used to accurately simulate the soil erosion processes in the Loess Plateau during the Holocene, see Zhao et al., (2022a, b). Their simulated soil erosion intensities are in good agreement with the geological evidence

of soil erosion in the Loess Plateau calculated from loess-paleosol profiles (Zhao et al., 2022a) and recorded by the sediment deposition rate in the Yellow River Delta (Zhao et al., 2022b). Our approach, which runs 5000 model years to get a steady-state topography is also widely used in other simulation works to get an initial topography (e.g. Campforts et al., 2020; Sharma et al., 2021). We have added the descriptions about the reasonability of the initial topography, please see them in the main text (line 273-278).

Line (273-278) This assumption is reasonable since recent studies have used the modern topography to accurately simulate the soil erosion processes in the Loess Plateau during the Holocene (Zhao et al., 2022a,b). Their simulated soil erosion intensities are in good agreement with the evidence provided from loess-paleosol profiles (Zhao et al., 2022a) and sediment deposition rates in the Yellow River Delta (Zhao et al., 2022b).

RC 11: Line 270: what is the method proposed by Chang et al. (2016)?

AR 11: Thanks for your comment. The method proposed by Chang et al. (2016) is the double-mass curves method (DMCs), which calculates the natural discharge and sediment load by analyzing the correlations between cumulative precipitation and annual discharge or sediment load. We have added the statement about this in the main text (line 285-290). The details about the method are described in the supplemental materials (Text S1, Table S3).

(Line 285-290) Since our models don't consider the impacts of e.g. dams and irrigation systems, mean annual discharge and sediment load data measured at the stations are re-calculated into natural discharge and sediment load data by using the double-mass curves method (DMCs). This method uses the correlation between cumulative precipitation and annual discharge or sediment load (Chang et al., 2016). Details are presented in the supplemental materials (Text S1, Table S3).

RC 12: Line 272-273: please show the acknowledged evaluation criterion and its results in a figure for the mentioned accepted calibration results.

AR 12: Thanks for your suggestions. We chose the evaluation criterion (the mismatches between the simulated and observed discharges and sediment loads are less than 10%) based on our previous simulation work in the Beiluo River (Chen et al., 2021), which is a tributary of the Wei River. This also applied in a work which used the SPACE model to study the effect of vegetation on soil erosion. In this previous work (Chen et al., 2021), the predicted errors in the hydrological stations are all around 10%. In the simulation works of Carriere et al. (2019), they used a "leave-one-out" calibration method based on a 23-year dataset. The satisfactory value of predicted error shown in their work is also around 10%. We have added the statement about why the evaluation criterion was chosen in the main text (line 292-295).

The accepted calibration results are shown in the Fig. S5 and Fig. S6, which are included in the Supplemental materials.

(Line 292-295) The calibration results are accepted when the mismatch between the simulated and observed discharges and sediment loads is less than 10%. This evaluation criterion was chosen based on the previous simulation works (Carriere et al., 2019; Chen et al., 2021).

(Supplemental Materials; Fig S5: The distributions of effective root depth of plants after calibration; Fig S6: The difference between simulated and observed sediment load (a), and the distributions of erodibility of the Base layer (b) and Surface layer (c) after calibration)

[Figure]

RC 13: Line 315: what are the climatic inputs for Normal and WCC scenarios, respectively? What is the temporal and spatial resolution of model outputs?

AR 13: Thanks for your questions. The climatic inputs for the Normal scenario are the reconstructed Holocene climate data, such as precipitation, air temperature, atmospheric CO2 concentration, etc. They are described in detail in Section 3.1.2 "Spatial distribution of climate data" (line 230-243). The WCC scenario used the same climate data from 6000 BC to 5500 BC as in the Normal scenario, and they are held constant through the whole simulation time. We had added a statement about the climatic inputs of the WCC scenario in the main text (line 340-342).

The spatial resolution of model outputs is 1000 m, and the temporal resolution is 1 year. However, we calculated the average value for each 500 years, since the annual climatic parameters and anthropogenic land cover are set constant for each 500 years interval. We have added descriptions about these in the main text (line 366-371).

(line 230-243) For the simulations of the Holocene (from 6000 BCE to AD 1850), the spatial distributions of climatic inputs are the same as those used in the calibration simulations. The Holocene precipitation and temperature series are calculated based on the annual precipitation reconstruction from Gonghai Lake (Chen et al., 2015) and the air temperature reconstruction from the Mangshan Loess plateau (Peterse et al., 2011). The methods are the same as those used in Chen et al. (2021). The calculated precipitation and air temperatures fit well with the reconstructed data from Beilianchi lake (Zhang et al., 2020, 2021) (Text S2, Fig. S2), which is located in the northwestern part of the Wei River catchment (Fig. 1a). Holocene atmospheric $CO_2$ concentrations are from the results of the Vostok ice core (Barnola et al., 1995; Petit et al., 1999). Holocene insolation values are calculated using the method of Laskar et al. (2004). The humidity and the sunshine duration values are set equal to modern values, because a sensitivity analysis has shown that variation of these two parameters has a limited impact on the results (Chen et al., 2021).

(line 340-342) Scenario WCC is used to study solely the effects of land use change; the climatic conditions are the same as those applied in the Scenario Normal from 6000 BCE to 5500 BCE, and they are kept constant during the simulation.

(line 366-371) In the Holocene simulation, the time-step is one year and the spatial resolution is 1000 m. Since the temporal resolutions of the reconstructed Holocene climate data, especially for the air temperature whose temporal resolution is around 500 years (Peterse et al., 2011), we set the annual climatic parameters and anthropogenic land cover constant for each 500 years interval, for simplicity. Therefore, the mean annual discharge or sediment yield are calculated for each 500 years.

RC 14: Line 345: why constant for each 500 years?

AR 14: Thanks for this question. The climatic inputs are set constant for each 500 years based on the temporal resolution of the reconstructed Holocene climate data, especially for the air temperature. The reconstruction of air temperature is from a loess-paleosol sequence at the Mangshan loess plateau (Peterse et al., 2011), the resolution is around 500 years. Therefore, we make the climatic input data constant for each 500 years interval to do the simulations. We have added statements in the main text to explain it (line 367-370).

> (line 367-370) Since the temporal resolutions of the reconstructed Holocene climate data, especially for the air temperature whose temporal resolution is around 500 years (Peterse et al., 2011), we set the annual climatic parameters and anthropogenic land cover constant for each 500 years interval, for simplicity.

RC 15: Line 354: what is the relation between runoff and discharge? how did authors calculate the discharge by runoff?

AR 15: Thanks for these questions. We calculated the discharge based on the simulated runoff by using the "FlowAccumulator" component in the Landlab model. The "FlowAccumulator" can calculate the routing of runoff, from hillslopes, via the river channels, to the outlet. We have added a technology roadmap of model development in section 3.1 to show more details about our model structure, as you suggested (line 188).

> (line188; Fig 2: Technology roadmap of the landscape evolution model.)

[Figure]

RC 16: From the difference of model results for the two scenarios, impacts of climate change can be identified but no evidence shows the effects of ALCC.

AR 16: Thanks for your comment. As you mentioned, the comparison between the two scenarios were used to show the direct impact of climate change. We modified the statements in the main text about the effects of ALCC based on the direct comparisons between the Normal and WCC scenarios (line 449-452). Then, we showed the effects of ALCC in the Section 5.1, which are represented by the changes in the sensitivity of discharge and sediment yield to climate change due to ALCC. Therefore, the effects of ALCC in our manuscript were not the direct impact on the discharge and sediment load, but the influence on the sensitivity of the response of discharge and sediment load to the climate change.

(line 449-452) The spatial variation coefficient of sediment yield in the two scenarios are almost the same during the simulation (less than 20%, Fig 5d), which indicates that climate change has limited impact on the spatial characteristics of sediment yield.

RC 17: In section 5.1, the sensitivity of discharge and sediment yield to the climate changes is based on statistical analysis rather than the ALCC impact on land-air interaction. It belongs to a complex climate system, so the title "Regime shift of a large river as a response to Holocene climate change depends on land use" is difficult to be addressed mechanically without climate-landscape evolution coupled modeling.

AR 17: Thanks for your comment. Our simulations didn't consider the impact of ALCC on the climate conditions, which is indeed an important way to affect the sensitivity of discharge and sediment yield to climate changes. The sensitivity change in our simulations is caused by the shift of the geographic center of land use change from the northwest to the southeast in the Wei River catchment. This shift made a switch of natural vegetation from grass to forest. Runoff in grassland is more sensitive to climate change than runoff in cropland, whereas runoff in forest is less sensitive than runoff in cropland. The switch of natural vegetation from grass to forest causes an abrupt change of the sensitivity. Then, we present a correlation between the spatial distribution of sediment accumulation and the distribution of archaeological sites during the mid-Holocene (Fig. 6). Based on this correlation, we put forward the possibility that the shift of the geographic center of land use change, which causes the change of sensitivity of the Wei River catchment to climate change, could be a result of the increase of the areal extent of land use. Therefore, we showed that the change of sensitivity of the Wei River catchment to climate change, which indicates a regime shift of the fluvial system would be caused by the increase of the areal extent of land use. We have made an explanation in the Section 5.1 and 5.2.

Our title "Regime shift of a large river as a response to Holocene climate change depends on land use" may cause misunderstandings. Therefore, we modified the title to "Past anthropogenic land use change causes a regime shift of the fluvial response to Holocene climate change in the Chinese Loess Plateau" to eliminate ambiguity.

---

## Author Comment (AC2)

**Past anthropogenic land use change caused a regime shift of the fluvial response to Holocene climate change in the Chinese Loess Plateau**

**Hao Chen[1, 2], Xianyan Wang[1*], Yanyan Yu[3], Huayu Lu[1] and Ronald Van Balen[2, 4*]**

[1]Frontiers Science Center for Critical Earth Material Cycling, School of Geography and Ocean Science, Nanjing University, Nanjing 210023, China.

[2]Department of Earth Sciences, VU University Amsterdam, Amsterdam 1081HV, The Netherlands.

[3]Key Laboratory of Cenozoic Geology and Environment, Institute of Geology and Geophysics, Chinese Academy of Sciences, Beijing 100029, China.

[4]TNO-Geological Survey of the Netherlands

*Correspondence to*: Xianyan Wang ([xianyanwang@nju.edu.cn](mailto:xianyanwang@nju.edu.cn)), Ronald Van Balen ([r.t.van.balen@vu.nl](mailto:r.t.van.balen@vu.nl))

**Summary:**

The authors would like to thank the Reviewer, Dr. Weimingliu, for the constructive and insightful comments on our manuscript, which were very useful for us to improve the manuscript and its readability.

We carefully revised the manuscript based on the Reviewer's suggestions. The modifications include the information about the KK10 model and the explanations about the evaluation criterion in the calibration processes. In the new version of the manuscript, the comments and suggestions raised by Reviewer are fully considered. We think the modified manuscript can meet reviewer's expectations.

In the following, we discuss in detail all Reviewer's comments and show how we have addressed them in the revised manuscript. Please note that the Reviewer's comments

are in black, our responses are in blue, and the content of the revised manuscript is depicted in a frame.

**Legend**

RC: Reviewer Comment;   AR: Author Response;   ☐: Modified manuscript content

**Response to the Referee Weimingliu #2:**

RC 1: Line 85: What is KK10? I think authors should add more information about it in the supporting information since it's maybe one of reasons for the sources of sensitivity.

AR 1: Thanks for your comments and suggestions. We have added a brief description about the KK10 scenarios in the Introduction (line 84-87). More information about KK10 model scenarios has been included in the Section 3.1.3 (line 245-257).

(line 84-87) The changes of anthropogenic land use are taken from Kaplan et al. (2010). Their KK10 database provides the anthropogenic land cover change from 8000 years ago to AD 1850, based on a model that relates changes of global population to past land use (Kaplan et al., 2009).

(line 245-257) The changes of anthropogenic land use since the mid-Holocene (Fig. S3) is obtained from the KK10 database, which in turn is calculated from a global ALCC model that is driven by population density and the land suitability (Kaplan et al., 2009, 2011). The land suitability takes into account that agriculture develops first on the most productive crop lands (Kaplan et al., 2009). The used time series of the KK10 model is from 6000 BCE to AD 1850. Because only the provincial data from 221 BCE to AD 1850 (Zhao and Xie, 1988) were available to calibrate the spatial patterns of population changes in China used by Kaplan et al. (2009), there is an uncertainty in the land-use changes in our study region prior to 221 BC. In previous simulations focusing on a tributary, the Beiluo River catchment, Chen et al. (2021) applied a variation of 25% for the ALCC from 6000 BCE to 221 BCE to estimate the impact of this uncertainty. It showed the uncertainty of ALCC had a limited effect on

the simulation results for discharges and sediment loads.

RC 2: Line 184: "model development" seems to be inaccurate in here, and I think "model summary" better sums up Sect 3.1.

AR 2: Thanks for your suggestion. We have changed the title of Sect.3.1 from the "Model development" to the "Model summary".

RC 3: Line 271: Is there any reason why the authors think that a 10% error is acceptable? In my image it is usually 5%.

AR 3: Thanks for your comment. We chose the 10% error based on our previous simulation work in the Beiluo River (Chen et al., 2021), which is a tributary of Wei River. This also applied in a SPACE model to study the effect of vegetation on soil erosion (Carriere et al., 2019). The predicted errors in the hydrological stations in the Beiluo River are all around 10% (Chen et al., 2021). In the simulation works of Carriere et al. (2019), they used a "leave-one-out" calibration method based on a 23-year dataset. The satisfactory value of predicted error shown in their works is also around 10%. We have added the statement about why the evaluation criterion was chosen in the main text (line 292-295).

(Line 292-295) The calibration results are accepted when the mismatch between the simulated and observed discharges and sediment loads is less than 10%. This evaluation criterion was chosen based on the previous simulation works (Carriere et al., 2019; Chen et al., 2021).

---

## Author Comment (AC3)

**Past anthropogenic land use change caused a regime shift of the fluvial response to Holocene climate change in the Chinese Loess Plateau**

**Hao Chen[1, 2], Xianyan Wang[1*], Yanyan Yu[3], Huayu Lu[1] and Ronald Van Balen[2, 4*]**

[1]Frontiers Science Center for Critical Earth Material Cycling, School of Geography and Ocean Science, Nanjing University, Nanjing 210023, China.

[2]Department of Earth Sciences, VU University Amsterdam, Amsterdam 1081HV, The Netherlands.

[3]Key Laboratory of Cenozoic Geology and Environment, Institute of Geology and Geophysics, Chinese Academy of Sciences, Beijing 100029, China.

[4]TNO-Geological Survey of the Netherlands

*Correspondence to*: Xianyan Wang ([xianyanwang@nju.edu.cn](mailto:xianyanwang@nju.edu.cn)), Ronald Van Balen ([r.t.van.balen@vu.nl](mailto:r.t.van.balen@vu.nl))

**Summary:**

The authors would like to thank the Reviewer, Dr. Amanda Schmidt, for the constructive and insightful comments on our manuscript, which were very helpful for us to significantly improve the manuscript and its readability.

Based on the Reviewer's suggestions, we carefully revised the manuscript to better present the processes of our modelling approach and to make readers easier understand the various steps in our simulations. We have added appropriate explanations for the setting of model parameters, detailed descriptions for the model structure and concise presentation for the model results. We also connect our works to other human-land use works in the Loess Plateau, as the reviewer suggested. Moreover, we modified the Discussion to better show the impact of the past anthropogenic land use change on the

sensitivity of the Wei River catchment to climate change. Issues in the figures and tables are resolved.

Overall, in the new version of the manuscript, the comments and suggestions raised by Reviewer are fully considered. We think that the modified manuscript can meet reviewer's expectations.

In the following, we discuss in detail all Reviewer's comments and show how we have addressed them in the revised manuscript. Please note that the Reviewer's comments are in black, our responses are in blue, and the content of the revised manuscript is depicted in a frame.

**Legend**

RC: Reviewer Comment; AR: Author Response; ☐: Modified manuscript content

RC 1: It isn't clear to me why some decisions for model parameters were made based on totally modern agricultural practices. The model requires soil nitrogen content and this was set as constant based on modern fertilization levels. I don't understand why this amount was chosen when fertilization would have varied over history. Similarly, the crop is assumed to be irrigated twice, but irrigation would have been different at different points in history. Finally, why was winter wheat chosen for all time steps? Do we know that that is what people grew in the past? If not, do we have data that could provide better information for past crop growth?

AR 1: Thanks for your comments and suggestions. As you mentioned, we set some model parameters (soil nitrogen content and irrigation strategy) the same as the modern agricultural practices. These parameters could be different at different points in history, but we didn't have enough data to determine the value of these parameters during the Holocene. Therefore, we assume they are the same as the modern values and keep them constant during the Holocene.

We chose the winter wheat for our Holocene simulations since it occurred in the middle reaches of Yellow River since about 3000 BCE (Dodson et al., 2013). The earlier type

of crop could be different, such as millet (Zhuang et al., 2014). However, we didn't have the data about the distributions of the millet during the Holocene in the Wei River. In addition, there are no millet's ecological parameters in previous studies that apply the Biome-BGC model, but they do have the winter wheat's ecological parameters (Hu et al., 2011). Therefore, we used the winter wheat as the crop during the Holocene simulations.

We have added a statement to explain the reason for setting these parameters in the main text (line 376-380).

(line 376-380) The type of crop and its management parameters, such as soil nitrogen content and irrigations, were set the same as the modern values, because of lack of available data. This assumption is reasonable because wheat has been cultivated in the middle reaches of Yellow River as early as the mid-Holocene (Dodson et al., 2013; Zhuang and Kidder, 2014).

RC 2: I had a hard time following all the different geographic names, locations, watersheds, gauging stations, and so on. I know these data are available in figure 1, but it is such important information and so hard to read in such a small figure. It might help readers who are less intimately familiar with the area to have a larger context map with the rivers, gauging stations, and sub-catchments clearly delineated.

AR 2: Thanks for your comment and suggestions. We have modified Figure 1 accordingly.

(Fig 1: The Wei River catchment a. Location of the Wei River and Yellow River; b. Landform types in the catchment; c. Meteorological stations, hydrological stations and rivers in and around the Wei River catchment.)

[Figure]

Meteorological station (Caliration) : 1. Minxian; 2. Yuzhong; 3. Huajialing; 4. Maiji; 5. Guyuan; 6.Kongtong; 7. Longxian 8. Taibai; 9. Fengxiang; 10. Huanxian; 11. Yanchi; 12. Changwu; 13. Wuqi; 14. Qindu; 15. Jingbian;16. Luochuan; 17. Pucheng; 18. Shangxian; 19. Huashan; 20. Yanchang

Meteorological station (Validation) : 1. Xiji; 2. Dingbian; 3. Xifeng; 4. Yongshou; 5. Wugong; 6.Yaoxian

Hydrogical station (Caliration) : 1. Wushan; 2. Qin'an; 3. Linjiacun; 4. Hongde; 5. Yangjiaping; 6.Yuluoping; 7. Qingyang 8. Jiaqiao; 9. Wuqi; 10. Zhangjiasha; 11. Xianyang; 12. Liujiahe; 13. Jiaokou; 14. Huaxian; 15. Zhuangtou

Hydrogical station (Validation) : 1. Beidao; 2. Jingchuan; 3. Weijiabao; 4. Banqiao; 5. Jingcun; 6.Lintong; 7. Huangling

RC 3: Several times in the results the authors have long lists of results that would be better presented in tables. It's very hard to follow long lists of results, especially for people who are less familiar with the study area.

AR 3: Thanks for your comment. We have added two tables (Table 2 and Table 3) to describe the total trend of discharge and sediment load in each sub-catchments from 6000 BCE to AD 1850, following your suggestions.

**Table 2 The total difference of mean annual discharge in each sub-catchment from 6000 BCE to AD 1850**

| Mainstream of Wei River | | Jing River | | Beiluo River | |
|---|---|---|---|---|---|
| Wushan | -13.2% | Hongde | -37.3% | Wuqi | -33.1% |
| Qin'an | -26.7% | Jiaqiao | -20.2% | Liujiahe | -29.6% |
| Linjiacun | -20.7% | Qinyang | -37.3% | Jiaokou | -39.9% |
| Xianyang | -31.1% | Yuluoping | -44.7% | Zhuangtou | -50.4% |
| Huaxian | -35.1% | Yangjiaping | -22.7% | | |
| Outlet | -21.8% | Zhangjiashan | -47.2% | | |

**Table 3 The total difference of mean annual sediment load in each sub-catchment from 6000 BCE to AD 1850**

| Mainstream of Wei River | | Jing River | | Beiluo River | |
|---|---|---|---|---|---|
| Wushan | +2519.9% | Hongde | +1275.3% | Wuqi | +1397.6% |
| Qin'an | +920.8% | Jiaqiao | +3100.4% | Liujiahe | +2407.5% |
| Linjiacun | +1687.5% | Qinyang | +1007.0% | Jiaokou | +1953.4% |
| Xianyang | +1506.7% | Yuluoping | +1282.1% | Zhuangtou | +1278.1% |
| Huaxian | +3126.9% | Yangjiaping | +1396.5% | | |
| Outlet | +906.8% | Zhangjiashan | +1411.1% | | |

RC 4: I'm concerned that the humidity is set to modern levels but we know that various times in the Holocene had very different humidity levels based on loess vs soil accumulation on the Chinese Loess Plateau. I wonder if this would then affect vegetation and erosion.

AR 4: Thanks for your comments and suggestions. Just as you mentioned, the humidity varied in the Wei River catchment during the Holocene. However, we set the humidity same as modern levels, since the data was lacking. Changes of humidity do have an impact on the absolute values of the simulated discharge and sediment load. We have done sensitivity analysis for the variation of humidity in the Beiluo River, which is a tributary of the Wei River, in our previous simulation works (Chen et al., 2021). The results show that 6% change in relative humidity could lead to a <17% variation of discharge and a <23% variation of sediment load. However, we found the variation of relative humidity had a limited impacts on the relative results between different scenarios (Chen et al., 2021). Therefore, the comparisons between different scenarios

are sufficiently accurate even though we set the relative humidity equal to the modern levels. We have added statements in the main text to explain the settings for the relative humidity in the Holocene simulations (line 240-243).

(line 240-243) The humidity and the sunshine duration values are set equal to modern values, because a sensitivity analysis has shown that variation of these two parameters has a limited impact on the results (Chen et al., 2021).

RC 5: Why was a 90 m DEM resampled (with, I assume, interpolation, although the exact interpolation was not specified) when there are high quality 30 m DEMs now available? It seems like an additional level of uncertainty that could have been avoided.

AR 5: Thanks for your comment. We also tried to resample the 30 m DEM (NASA SRTM Version 3.0 Global 1 arc second dataset) to a resolution of 1000 m. However, we found there were more artefacts in the resampled DEM from 30 m than that from 90 m. The artefacts make the river network disrupted when we perform the simulations. In order to guarantee the success of our simulation works, we chose to use the NASA SRTM 90m DEM.

RC 6: The authors talk a lot about base layers and surface layers but never specify what they are using for base and surface layers. Is the base layer loess? If so, how is loess thickness set when loess is continuously being deposited? Regardless, it would be good to explicitly say what the base and surface layers are composed of.

AR 6: Thanks for your comment and suggestion. In our simulations, the Surface layer is composed of sediment produced by hillslope- and fluvial process. The Base layer is composed of bedrock, which we obtain from the geological map (Fig S4a). Therefore, for the area covered by loess, the Base layer is loess. We didn't consider the deposition of loess during our simulations. We have added statements about the materials of Base and Surface layers to make a clear description (line 259-264).

(line 259-264) There are two layers in the landscape evolution model, a Base layer and a Surface layer (Shobe et al., 2017). The Surface layer consists of loose material

and is above the Base layer, which is composed of basement (i.e. bedrock and loess in different areas). The Surface layer is composed of sediment produced by hillslope- and fluvial processes. The material of Base layer is set based on the rocky types, consisting of loess, sandstone, etc. (Fig S4a).

RC 7: What is the correction used for dams and irrigation channels? How do we know that is reasonable in the past?

AR 7: Thanks for your comment. We didn't consider the effect of dams and irrigation projects in our models. The human activities in our simulations are land use changes. In our calibration simulations (from 1996 to 2016), we first re-calculate the observed mean annual discharge and sediment load data collected from hydrological stations to the natural mean discharge and sediment load data, which are not affected by dams and irrigation systems. Then, we used the natural mean discharge and sediment load data to calibrate the parameters in our landscape model. Therefore, the results of Holocene simulations are the hydrologic and sediment processes without the effects of dams and irrigation systems. We have added some statement about these in the main text (line 285-288)

(line 285-288) Since our models don't consider the impacts of e.g. dams and irrigation systems, mean annual discharge and sediment load data measured at the stations are re-calculated into natural discharge and sediment load data by using the double-mass curves method (DMCs).

RC 8: All the web resources the authors list (mostly data sources) should be properly cited rather than just webpages listed in the text.

AR 8: Thanks for your suggestion. We have replaced the webpages with appropriate citations in the main text.

RC 9: In the discussion, I wasn't convinced by the argument that climate change is less important than human activity. It needs more explanation.

AR 9: Thanks for your comments and suggestions. Because the comparisons between Normal and WCC scenarios showed that the changes of climate variations would cause a significant **decrease** for both mean annual discharge and sediment yield after 1000 BCE (Fig. 5), we attributed the predicted significant **increase** of sediment thickness and sediment yield mainly to the large increase of land use rather than climate change. The contributions of climate change to the mean annual discharge and sediment yield do have some increases after 1000 BCE (Fig. 5). We think they are signals indicating the change of sensitivity of the fluvial catchment to climate change as a result of increasing ALCC. We have added some new statements in the main text to make a clearer explanation line (461-465).

> (line 461-465) Since the comparisons between Normal and WCC scenarios showed the changes of climate variations would cause a significant decrease for both mean annual discharge and sediment yield after 1000 BCE (Fig. 5), the increments of sediment thickness and sediment yield should be a consequence of the large increase of the land use around the 1000 BCE.

RC 10: Likewise, I didn't understand the argument about the multiple thresholds of sensitivity to change. This seems interesting but was not well enough explained.

AR 10: Thanks for your comment. In Section 5.1, we first calculated the changes in the sensitivity of discharge and sediment yield to climate change. We used the abrupt changes of sensitivity as the marker of the regime shift in the Wei River catchment. Then, we attributed the abrupt change of sensitivity to the shift of the geographic center of land use change from the northwest to the southeast of the Wei River catchment, which made a switch of natural vegetation from grass to forest. The runoff in grassland is more sensitive to climate change than runoff in cropland, whereas runoff in forest is less sensitive than runoff in cropland. The switch of natural vegetation from grass to forest would make the sensitivity to climate change from decrease to increase. In Section 5.2, we present a correlation between the spatial distribution of sediment accumulation and the distribution of archaeological sites during the mid-Holocene (Fig.

6). Based on this correlation, we put forward the possibility that the shift of the geographic center of land use change, which causes the change of sensitivity of the Wei River catchment to climate change, could be a result of the increase of the areal extent of land use. Therefore, we showed that the change of sensitivity of the Wei River catchment to climate change, which indicates a regime shift of the fluvial system, would be caused by the increase of the areal extent of land use and also the exact changes of the kinds of vegetation (i.e from grass or forest to cropland). We have modified the statements in the Discussion to make the explanation clearer.

RC 11: It would be interesting to see this work connected to other human-land use work in China. There is such a rich literature on this topic, including from lake cores, simulations, and geochemistry. This paper would have a much bigger impact if it drew those connections more explicitly.

AR 11: Thanks for your comment and suggestion. We have compared our simulation works with other human-land use work in the Loess Plateau in the main text (line 465-476).

(line 465-476) This is in agreement with the evaluation of human-land use change accelerated soil erosion in the Loess Plateau recorded by colluvial components in Holocene loess–soil sequences (Huang et al., 2006), dike breaches (He et al., 2006), temporal changes of sedimentation rate from the Yellow River delta (Zhao et al., 2013) and previous modelling of the Beiluo River tributary of the Yellow River (Chen et al., 2021). The onset of the human-dominant soil erosion in our simulations (~1000 BCE) could be earlier than the inferences from the sedimentation rate records in Beilianchi lake (see the location in Fig. 1a; ~ AD 600; Zhang et al., 2019) and simulated soil erosion rate in the middle reaches of Yellow River (~ AD 1; Zhao et al., 2022a,b). These differences may be caused by the spatial variations of the development of agriculture in the Loess Plateau (Zhuang and Kidder, 2014; Yu et al., 2016).

RC 12: There are a few typos in figures and tables that need to be fixed. For example, table 1 has ages for Yangshao that don't overlap the time period for that row. Figure 5c has a typo in the title. In general, figures that have small text and a mix of red to green colors can be hard for many people to read.

AR 12: Thanks for your comment. All the typos in figures and tables have been corrected. The small text and a mix of red to green colors in figures also have been modified.